



# A regional evaluation of the influence of climate change on long term trends in chlorophyll-a in large Italian lakes from satellite data

Gary Free[1], Mariano Bresciani[1], Monica Pinardi[1], Nicola Ghirardi[1], Giulia Luciani[2], Rossana Caroni[3] and Claudia Giardino[1]

[1] Institute for Electromagnetic Sensing of the Environment, National Research Council, Milan, 20133, Italy
[2]Department of Civil and Environmental Engineering, Politecnico di Milano, Milan, Italy
[3]Water Research Institute, National Research Council, Verbania, Pallanza, Italy

*Correspondence to*: Gary Free (gnfree@hotmail.com)

**Abstract.** Climate change has increased the temperature and altered the mixing regime of high-value lakes in the sub-alpine region of Northern Italy. Remote sensing of chlorophyll-a can help provide a time-series to allow an assessment of the ecological implications of this. Non-parametric multiplicative regression (NPMR) was used to visualize and understand the changes that have occurred between 2003-2018 in lakes Garda, Como, Iseo and Maggiore. In all four deep sub-alpine lakes there has been a disruption from a traditional pattern of a significant spring chlorophyll-a peak followed by a clear water phase and summer/autumn peaks. This was replaced after 2010-2012, with lower spring peaks and a tendency for annual maxima to occur in summer. There was a tendency for this switch to be interspersed by a two-year period of low chlorophyll-a, which seemed to extend until 2018 for Lake Garda. Variables that were significant in NPMR included time, air temperature, wind speed, cloud cover, winter temperature and winter values for the North Atlantic Oscillation and Eastern Atlantic pattern. The change from spring to summer chlorophyll-a maxima, relatively sudden in an ecological context, could be interpreted as a regime shift. The cause is probably cascading effects from increased winter temperatures, reduced winter mixing and altered nutrient dynamics. Future trends will depend on climate change and inter-decadal climate drivers.

## 1 Introduction

Lakes are a useful medium through which to see the effects of climate change. Their nature as a semi-contained system – catchment and lake basin with distinct and fast moving seasonal succession of flora and fauna, especially in the plankton, marks them as rapidly responding ecosystems ideally suited to observe change. While the response may be complex resulting from interacting pressures, lake typology and position on the trophic gradient, there is also significant value in interpreting an integrated signal to understand the impact in the natural environment (Adrian et al., 2009; Weyhenmeyer et al., 1999).

Climate change is an important driver of global biodiversity in lakes and has been ranked third after invasive species and land-use change (Sala et al., 2000). Projections suggest that climate change will have a much more widespread and substantial effect in coming decades altering land cover, hydrology, nutrient cycling and species composition (Cardoso et al., 2009; Carpenter et al., 2011).

The global increase in summer surface temperature has been estimated as 0.34°C decade[-1] but the influence of local and lake specific parameters such as morphology mean that there is not often regional consistency to these trends (O'Reilly et al., 2015). In addition, many lakes in Europe now no longer cool to 4ºC, the maximum density of water. This critically affects a key mechanism, colder water sinking, by which deep lakes mix (Woolway et al., 2019). The ecological effects of climate change are complex, lake specific and often depend on multiple interacting pressures (Craig et al., 2017). For example, phytoplankton can undergo changes to species composition, production, size structure and phenology which are likely to result in (and from) significant alteration to ecosystem functions at several trophic levels (Jeppesen et al., 2014;





Weyhenmeyer et al., 1999; Winder and Sommer, 2012). Higher temperatures in summer and more stable stratification has been predicted to lead to increased harmful algal blooms (O'Neil et al., 2012).

The Italian sub-alpine lakes are of key economic importance and their size and depth makes them a key regional water resource requiring priority management (Premazzi et al., 2003; Regione del Veneto, 2018). A warming trend has been detected in the lakes with annual average surface temperatures increasing 0.017 °C yr⁻¹ and 0.032 °C yr⁻¹ in summer (Pareeth

et al., 2017). This has led to more stable stratification in the lakes increasing the isolation of the lower layers (hypolimnion) from the upper layers (epilimnion) with no complete mixing since 2006. This reduced mixing has led to a decreasing trend in oxygen concentrations in the hypolimnion with the result that climate now exerts more control on oxygen than trophic status in these lakes (Rogora et al., 2018). This has also reduced nutrient transfer from the hypolimnion to the epilimnion as documented for Lake Garda resulting in alterations to phytoplankton composition (Salmaso et al., 2018). Most studies have

attributed the cause to long term climate change and fluctuations in large scale regional climate drivers such as the North Atlantic Oscillation (NAO) and especially the East Atlantic pattern (EA) during winter (Rogora et al., 2018; Salmaso et al., 2018).

In this context, satellite remote sensing can enable the observation of a suite of functionally relevant indicators of water quality and ecosystem condition for the study of long-term environmental trends in lakes (Tyler et al., 2016). A bibliometric

analysis (Topp et al., 2020) has shown how optical remote sensing of lakes has expanded substantially over the past 50 years, with a sharp increase in the past 10–15 years as a consequence of improvements in sensor resolution. In this context, one of the most studied parameters is phytoplankton and in particular chlorophyll-a as a proxy of phytoplankton biomass. The spatial coverage and temporal sampling frequency achievable with satellite images absolutely provide novel insights into phytoplankton dynamic processes in lakes (e.g. Ho et al., 2019) that cannot be easily captured through in situ sampling

(Palmer et al., 2015). In Europe, remote sensing has been identified as a key component of future implementation of the Water Framework Directive (WFD) that requires lakes are ecologically assessed using biological quality elements including phytoplankton (Carvalho et al., 2019; Council of the European Communities, 2000, 2013).


As regards the use of satellite observations of phytoplankton in sub-alpine lakes, several studies have been published since the mid-90s (Zilioli et al., 1994) for developing and testing algorithms to retrieve chlorophyll-a from space (e.g. Giardino et al., 2014 and references in Odermatt et al., 2010), whether for assessing chlorophyll-a changes in single lakes (Odermatt et

al., 2008) or for the entire sub-alpine lake district (e.g. Bresciani et al., 2011 and reference therein). Specific studies have also been performed for assessing phytoplankton growth as a consequence of Saharan dust depositions (Di Nicolantonio et al., 2015), detecting cyanobacteria blooms (Bresciani et al., 2011c, 2018), or to analyze phytoplankton dynamics in relation to lake surface water temperature (Bouffard et al., 2018; Bresciani et al., 2011b).

Here we examine a 16 year (2003-2018) timeseries for four deep sub-alpine Italian lakes in order to identify changes in chlorophyll-a in the context of climate change. Satellite observations of chlorophyll-a provide higher temporal resolution that in addition to their capacity to integrate signals over a large spatial area of the lake allow a synoptic standardized picture to be leveraged to examine regional changes. We expect that the lack of stratification since 2005/2006 should be manifested in a disruption to the ecological functioning of the lake visible in altered chlorophyll-a seasonal patterns and concentration.




## 2 Methods

### 2.1 Study sites

The four lakes are situated in the sub-alpine region of northern Italy and are all large (Figure 1, Garda: 368 km$^2$, Como: 145 km$^2$, Iseo: 61 km$^2$, and Maggiore: 212 km$^2$). Owing to climatic conditions and the deep maximum depth of the lakes (Garda:

350 m, Como: 410 m, Iseo: 258 m, and Maggiore: 370 m) mixing historically only occurred every 7 years, typically down to a depth of 200 m during cold and windy winters, while mixing in Iseo is prevented by a salinity gradient. The last mixing event was during 2005/2006 (Leoni et al., 2019; Rogora et al., 2018; Salmaso et al., 2018). Total phosphorus trends have been analysed from 1992-2016 (Rogora et al., 2018). Trends and recent (2016) concentrations were reported as increasing in Iseo (90 µg l$^{-1}$) and Maggiore (13 µg l$^{-1}$) and declining in Garda (18 µg l$^{-1}$) and Como (35 µg l$^{-1}$) for spring whole-column

averages. However, the increases in Maggiore and Iseo were attributed to an increase in hypolimnetic concentrations whereas epilimnion (0-20 m) concentrations have been relatively stable after the 2005/2006 mixing event.

### 2.2 Collection and processing of satellite data

The satellite measurements of chlorophyll-a concentrations used in this study were obtained from four optical sensors that

allowed us to cover the temporal range 2003-2011 and 2014-2018. The older set of data was provided by MERIS (Medium Resolution Imaging Spectrometer) onboard of the Envisat-1 platform for nine years and offered imagery data with a spatial resolution of 300 m. The more recent data-set was built with images acquired by optical sensors OLI (Operational Land Imager), MSI (multi-spectral instrument) and OLCI (Ocean and Land Colour Instrument), onboard Landsat-8, Sentinel-2 and Sentinel-3 respectively. The multispectral MSI sensor provided imagery data with a geometric resolution of 10 m to 60 m,

the OLI sensor was 30 m, while the OLCI data offered a 300 m pixel resolution. Considering that the twin Sentinel-2 and Sentinel-3 satellites have been operational since 2015 (5 days revisit time) and 2017 (1-2 days revisit time) respectively, and adding the 16-day revisiting time of Landsat-8 as well as the fact that some lakes can be revisited twice due to the overlap of adjacent paths there is at last an adequate replacement of MERIS which had a 2-3 days revisit time.

For each sensor, satellite observation of chlorophyll-a data were obtained by processing cloud-free at-satellite-radiance data corrected for radiometric noise (e.g. atmospheric effects) to allow the computing of chlorophyll-a from atmospherically corrected images. In particular, for MERIS data the processing was based on the method described in Odermatt et al., (2010), which has been successfully adopted and validated in the study area (Bresciani et al., 2011a; Giardino et al., 2014b). In brief, radiometric noise was corrected using the BEAM-VISAT software (SMILE correction) (Fomferra and Brockmann, 2006),

adjacency effects using the tool 'Improved Contrast between Ocean and Land' (Santer and Schmechtig, 2000), and for simultaneously performing the correction of atmospheric effects and the retrieval of chlorophyll-a the 'Case-2 Regional neural network processor' was used (Doerffer and Schiller, 2008b, 2008a).

In the case of newer OLI, MSI and OLCI imagery, data have been processed using validated methods (Bresciani et al., 2018; Cazzaniga et al., 2019). Briefly, the radiative transfer 6SV code (Vermote et al., 1997) was used to first correct imagery data

from atmospheric effects. Chlorophyll-a was then retrieved using the tool BOMBER (Giardino et al., 2012), which implements a bio-optical model that was parametrised with the specific inherent optical properties of subalpine lakes (Bresciani et al., 2018; Giardino et al., 2014a; Manzo et al., 2015).

For each chlorophyll-a product inferred from MERIS, OLI, MSI and OLCI, regions of interest (ROI) were defined to extract the average value and standard deviation. The ROIs were defined to cover the equivalent surface area in units of km$^2$ so as to

be not dependent on pixel size, which varies across the sensors. The ROIs were distributed in different areas of the lake,



including their sub-basins and pelagic waters, while a buffer of 1 km from the land was used to exclude the issues of shallow waters, mixed pixels and residual noise for having miss-corrected adjacency effects.

### 2.3 Data compilation and treatment

After compiling the chlorophyll-a data it was evident that there were several gaps in the dataset owing to image unsuitability, often caused by cloud coverage, or in the case of 2012 the failure of the MERIS sensor satellite. The first approach to filling these gaps was to use chlorophyll-a data collected by regional authorities (ARPA Lombardia, ARPA Veneto) or using published values where available (Fenocchi et al., 2019). Satellite data was used preferentially to monthly in situ values in order to promote consistency in the dataset. In situ data were typically higher than that estimated by satellite and were down weighted using regression (the median value down weighted by was 1 µg l⁻¹). Where gaps remained, the missing values were

imputed using the package imputeTS in R (Moritz and Bartz-Beielstein, 2017; R Core Team, 2019). The dataset over the 192 months (2003-2018) comprised between 56-68% satellite data with in situ and imputed ranging from 8-31% (Table 1). Despite the missing months, the total satellite observations were in excess of the months included in this study – giving an indication of the relatively high frequency obtained compared to traditional monitoring (Table 1). The process of averaging

several values to give a monthly average will effectively smooth the data relative to single values that may or may not capture bloom events. In order to test whether the imputed values were over influencing the results, the statistical analysis for Lake Como, with the highest percentage of imputed values (31%) was rerun with imputed values deleted. The resulting output contour plot was very similar to that obtained on the complete data set with the same pattern of inversion of seasonal pattern of chlorophyll-a (described latter and reported in supplementary material: Fig. S1). Total phosphorus data was

obtained from regional authorities (ARPA Lombardia, ARPA Veneto) and published values (Leoni et al., 2019; Rogora et al., 2018).

The NAO and EA values were obtained from NOAA-CPC (https://www.cpc.ncep.noaa.gov/products/precip/CWlink/pna/nao.shtml). Climatic date were obtained from ERA5 - the fifth generation ECMWF reanalysis for the global climate and weather

(https://cds.climate.copernicus.eu/cdsapp#!/home). Data used for analysis included wind vectors (also expressed as speed and direction), total precipitation, air temperature, cloud cover and humidity. Winter average values (DJF: December, January, February) were also calculated for NAO, EA and air temperature as this has previously been found to be a key controlling parameter for these lakes (Leoni et al., 2018; Rogora et al., 2018; Salmaso et al., 2018). The approach to analysis, primally based on satellite and climate data compiled and available at global level should also ensure that the approach is

scalable and portable to other areas, an important criterion for global change assessments.

**Table 1.** The percentage of data used in analysis derived from satellite, in situ and imputed for the 192 months (2003-2018). In calculating monthly averages, often there were several satellite estimates per month available - visible in the total satellite observations column.

| Lake | Sat obs % | In situ % | Imputed % | Total sat. obs. |
|---|---|---|---|---|
| Maggiore | 68 | 9 | 23 | 308 |
| Garda | 56 | 24 | 20 | 311 |
| Como | 61 | 8 | 31 | 277 |
| Iseo | 57 | 15 | 28 | 270 |






### 2.4 Statistical analysis

Nonparametric Multiplicative Regression (NPMR) (McCune, 2006) was used to estimate the response of chlorophyll-a to climate and environmental parameters listed above. NPMR can define response surfaces using predictors in a multiplicative rather than in an additive way. This method is progressive in better defining unimodal responses more typical of ecological data than other methods such as multiple regression (McCune, 2006). It has previously been applied to model tree species distribution (Yost, 2008), the response of lichens to climate change (Ellis et al., 2007) and in time-series analysis (Nicolaou

and Constandinou, 2016). NPMR was applied using the software HyperNiche version 2.3 (McCune and Mefford, 2009). The response of chlorophyll-a was estimated using a local mean multiplicative smoothing function with uniform weighting. NPMR models were produced by adding predictors stepwise with fit expressed as a cross-validated $R^2$ ($xR^2$) which can be interpreted in a similar way as a measure of fit as a traditional $R^2$. NPMR models were evaluated using a Monte Carlo procedure where concentration was randomised, the procedure rerun, and the proportion of models (p) (with the same

number of predictors) with an $xR^2$ greater than or equal to the original model evaluated. The sensitivity, a measure of influence of each parameter included in the NPMR model, was estimated by altering the range of predictors by +/-0.05 (i.e. 5%) with resulting deviations scaled as a proportion of the observed range of the response variable. Therefore a value of 1 would correspond to change of equal magnitude in response and predictor variables. Sensitivity can be used to evaluate the relative importance of variables included in models because NPMR models are unlike linear regression and have no fixed

coefficients or slopes. The order of inclusion of variables in models is less important. Instead of fitting coefficients in NPMR tolerances are fit. These are reported in original units and represent one standard deviation of the smoothing function. Kendall's tau test and the Theil-Sen slope for the significance and estimate of time series trends was carried out on environmental data in R (Jassby and Cloern, 2014).


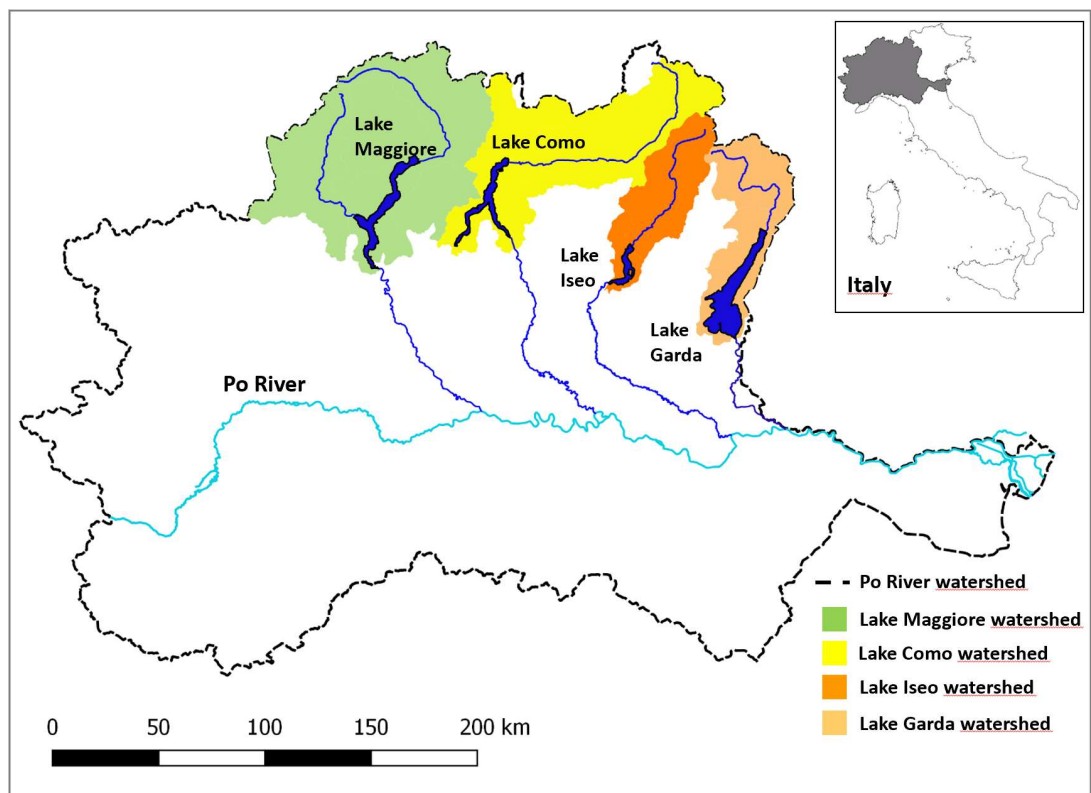

**Figure 1:** Location of the four sub-alpine lakes in Northern Italy.

### 3 Results

Chlorophyll-a in the four lakes showed significant variation over the 2003-2018 period (Fig. 2). Up until 2010 (Como, Iseo) or 2012 (Garda, Maggiore) the seasonal pattern was typically defined by a significant winter/spring (December-March) bloom followed by a clear water phase and summer/autumn peaks. This was followed by a two year period of low chlorophyll-a in lakes Como and Iseo (2010-2011), Maggiore (2012-2013) and from 2012 an extended period of lower values in Garda. This was followed by the remaining period up to 2018 where the seasonal pattern had less defined spring

blooms and a tendency for annual maxima to occur in summer. Examining satellite data alone the individual observations allow some higher peaks to be visualized that may otherwise be smoothed through monthly averaging (Fig. 3). In addition, gaps are visible where no suitable images were available, including a large gap where the MERIS sensor satellite was not available in 2012. Nonetheless, the aforementioned transition in seasonal patterns is still visible changing from essentially a concave to a convex annual pattern with more prominent mid-year concentrations.


   In order to better understand the variation in chlorophyll-a over time we carried out a non-parametric multiplicative regression analysis on each of the lakes (Table 2). The models for all four lakes were significant with $xR^2$ ranging from 0.44 to 0.59. Time and air temperature were consistently included in models for all four lakes. Several variables averaged over the winter period were also included in models for all lakes: temperature (Garda) DJF NAO (Maggiore) or DJF EA (Como,

Iseo). Wind speed was included in three models and cloud cover in one. The sensitivity value provides an indication of the importance of the variables in the models. In Lake Garda winter temperature had the highest sensitivity value (0.37) and in





Iseo the winter EA was highest (0.24). In Maggiore and Como the time variable had the highest sensitivity (0.39, 0.41 respectively). Temperature sensitivity values ranged from 0.17 – 0.32 among the four lakes. Wind had either the lowest or joint lowest sensitivity values when included in models (0.07-0.17).


Contour plots were produced to visualize model results for chlorophyll-a with temperature over time (Fig. 4). The contour lines and colour intensity represent chlorophyll-a in 1 µg l$^{-1}$ increments. Common to all the lakes are the higher concentrations in the bottom left of the figures occurring at lower temperatures (<10 °C) typically found in winter-spring during the period 2003-2010. Directly above this in the upper left are lower concentrations typically found at higher summer

temperatures during this period. This pattern then appears to change. From about 2010 or 2013 the pattern is inverted with higher concentrations now being found at higher temperatures (summer) and lower concentrations at lower temperatures in the winter-spring. Garda appears to be an exception with a transition to lower, almost uniform, concentrations at all temperatures.

The directional influence of the other variables was examined through additional NPMR response plots (see supplement). More negative values of the climate index DJF NAO in Lake Maggiore were associated with higher estimated chlorophyll-a concentrations, especially after 2009 (Fig. S2). Negative values of the DJF EA in Lake Como also were associated with higher chlorophyll-a up until 2015 when the reverse was true albeit with minor influence on chlorophyll-a (Fig. S3). Similarly, in Iseo, negative values of the DJF EA less than -1.5 were associated with higher chlorophyll (Fig. S4). In Lake

Garda higher chlorophyll-a was associated with DJF air temperatures below 4 °C, especially before 2010 (Fig. S5). Winter air temperature was positively correlated with winter EA for Como ($r_S = 0.82$, $p < 0.01$), Iseo ($r_S = 0.69$, $p < 0.01$) and Garda ($r_S = 0.58$, $p = 0.02$) but not Maggiore ($r_S = 0.42$, $p = 0.11$) where it was more correlated with the NAO ($r_S = 0.68$, $p < 0.01$).

Wind speed was included in the model for Como but the response plot (Fig. S3) indicated a changing response over time.

Chlorophyll-a was higher with higher winds earlier in the dataset but after 2010 lower wind speed, typical of summer, was associated with higher chlorophyll-a. The same pattern was observed for Iseo, although there was also a tendency for chlorophyll-a to be higher at wind speeds averaging 0.46 m s$^{-1}$ per month (Fig. S4). In Garda, windspeed higher than 0.6 m s$^{-1}$ per month was associated with higher chlorophyll-a until 2010 after which little response in phytoplankton biomass was noted (Fig. S5).


In order to further examine if the peak of chlorophyll-a has moved over time the maximum was calculated between June to May the following year - traditionally trough to trough (Fig.2, Fig. 5). The air temperature from when the maximum occurred was used to indicate disruption to the seasonal pattern over time. Evidence for significant increase was found for Como (sen slope = 0.94, p = 0.01) and Iseo (sen slope = 0.70, p = 0.01) but not Maggiore (sen slope = 0.85, p = 0.06) or

Garda (sen slope = 0.35, p = 0.49) the latter of which displayed a very limited seasonal pattern in chlorophyll-a since 2012.

To indicate environmental change over time that may be contributing to the disruption and alteration of chlorophyll-a concentrations we tested trends in a longer dataset (1980-2018) for the four lakes (Table 3). Trends in air temperature for all four lakes were highly significant with slopes ranging from 0.062 to 0.069, similarly DFJ air temperature slopes ranged from

0.049 to 0.073. The total increase in annual average ranged from 2.3 to 2.6 °C over the 39 years. Rainfall and wind speed did not show a significant difference in annual averages with the exception of Lake Iseo which had a slight decline in wind speed (Table 3).







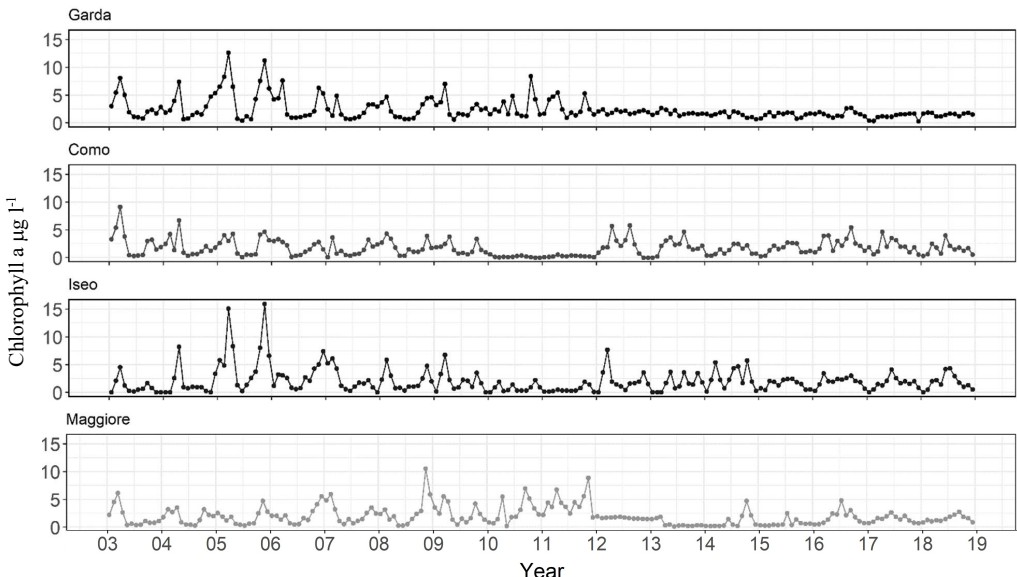

**Figure 2:** Chlorophyll-a in four subalpine lakes from 2003-2018 (satellite, in situ and imputed).


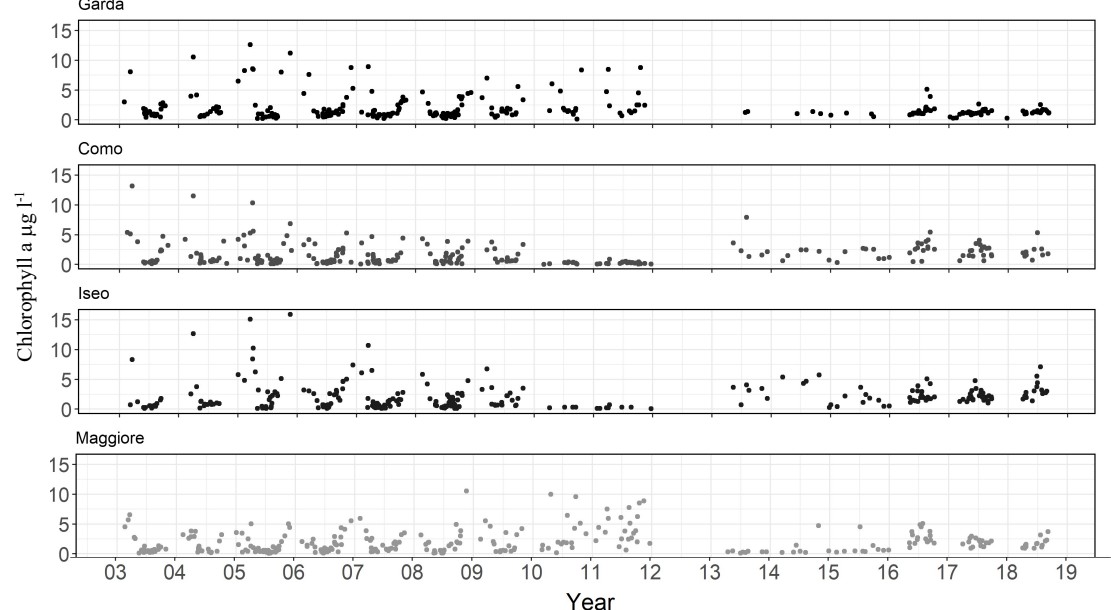


**Figure 3:** Chlorophyll-a in four subalpine lakes from 2003-2018 (satellite only).

**Table 2.** Results of NPMR (Non-Parametric Multiplicative Regression) models for Chlorophyll-a (Chl-a; n = 192, 2003-2018). $xR^2$ = cross-validated $R^2$; Ave. size = Average neighbourhood size; Tol. = Tolerance; Sen. = Sensitivity. Cloud co. = cloud cover; DJF_°C = December, January February mean temperature; DJF_EA = December, January February mean Eastern Atlantic values; DJF_NAO = December, January February mean North Atlantic Oscillation values; Wind = wind speed. Parameters with the highest sensitivity value for each lake are in bold.

| Lake Chl-a | $xR^2$ | Ave. Size | Variable 1 | Tol. | Sen. | Variable 2 | Tol. | Sen. | Variable 3 | Tol. | Sen. | Variable 4 | Tol. | Sen. |
|---|---|---|---|---|---|---|---|---|---|---|---|---|---|---|
| Garda | 0.59 | 9.80 | Time | 49.66 | 0.13 | Wind | 0.48 | 0.07 | °C | 7.24 | 0.20 | DJF_°C | 0.53 | **0.37** |
| Como | 0.44 | 11.40 | Time | 34.38 | **0.41** | Wind | 0.47 | 0.14 | °C | 4.64 | 0.32 | DJF_EA | 1.89 | 0.16 |
| Iseo | 0.55 | 9.90 | Time | 80.22 | 0.17 | Wind | 0.29 | 0.17 | °C | 7.03 | 0.17 | DJF_EA | 0.47 | **0.24** |
| Maggiore | 0.45 | 11.80 | Time | 32.47 | **0.39** | Cloud | 0.06 | 0.22 | °C | 4.73 | 0.27 | DJF_NAO | 1.77 | 0.07 |






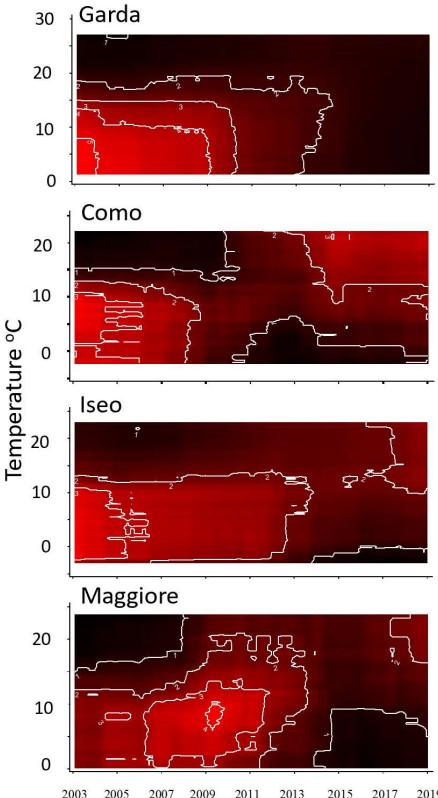

**Figure 4:** Model results (see Table 2) for chlorophyll-a μg l$^{-1}$ (contour lines) against year and °C.


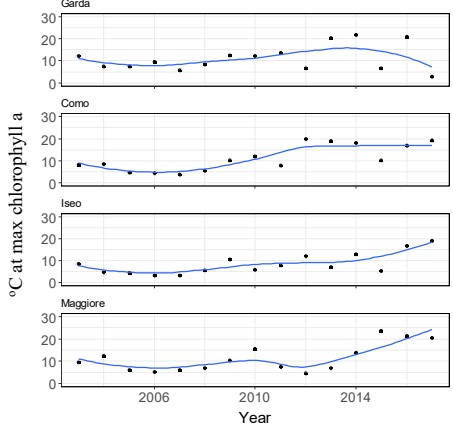

**Figure 5:** The temperature at which the maximum value of chlorophyll-a occurred for the four lakes from 2003 until 2017. Maximum values were calculated from June to May the following year. Loess smoother fitted to data.




**Table 3** Sen slope results on annual data from 1980-2018. *** = p ≤ 0.001, ** = ≤ 0.01.

| Parameter | Garda | Maggiore | Iseo | Como |
|---|---|---|---|---|
| ºC | 0.062*** | 0.062*** | 0.069*** | 0.069*** |
| DJFºC | 0.049** | 0.054*** | 0.073*** | 0.063*** |
| Wind speed | 0.000 | 0.001 | -0.002** | -0.001 |
| Rain | 0.000 | 0.000 | 0.000 | 0.000 |

**4 Discussion**

The dataset covered four lakes across the Italian sub-alpine region of differing trophic condition. Examination of the dataset and NPMR model contour plots indicated that there has been a disruption to the pre 2010 pattern of well-defined spring peak, clear water phase followed by summer peaks. Following a two year period of low chlorophyll-a (which extended for Lake Garda) the seasonal pattern had less defined spring blooms and a tendency for annual maxima to occur in summer.

Contour plots from the NMPR analysis indicated that this pattern of shifting spring to summer maxima was similar for Como, Iseo and Maggiore. The consistency of this pattern of change suggests a regional controlling influence such as climate rather than catchment specific factors.

All of the NPMR models included a winter climatic parameter (NAO, EA, winter air temperature) indicating the importance

of regional climate patterns in controlling chlorophyll-a. Lakes Como and Iseo were predicted to have higher chlorophyll-a concentrations when the winter EA was more negative. Similarly a colder winter air temperature was associated with higher chlorophyll-a in Lake Garda. A more negative EA is associated with colder air temperatures (Salmaso et al., 2018) and a significant correlation was found for lakes Como, Iseo and Garda in this dataset. In contrast, for Lake Maggiore, the most westerly lake, the winter air temperature was only correlated with the winter NAO which was included in the model and

indicated higher chlorophyll-a concentrations at lower values of the NAO. The importance of winter climate variables was further underlined by them being the most sensitive parameters included in NPMR models for lakes Garda and Iseo.

The destabilisation of stratification with lower surface water temperatures and sinking of colder water in winter is one of the principal mechanisms contributing to deep lake mixing (Horne and Goldman, 1994; Woolway et al., 2019) and positive

values in the EA and NAO, resulting in higher winter temperatures are likely to critically alter this process. The winter EA has been found to be increasing since the 1950s (Salmaso et al., 2018). Recent analysis on 635 lakes worldwide has predicted that for 100 lakes their mixing regimes are likely to be altered under climate change scenarios with some at risk of becoming permanently stratified, reducing nutrient upwelling and decreasing hypolimnion oxygen (Woolway and Merchant, 2019). This may be the situation for this set of sub-alpine lakes as positive values in the EA and NAO, and higher winter

temperatures have been linked to a shallower mixing depth and the failure of the Italian sub-alpine lakes to fully mix since 2005/2006 (Rogora et al., 2018). The observed decline in spring chlorophyll-a and alteration of the seasonal pattern is therefore likely a response to lower transport of nutrients to the epilimnion and the altered physical structure imposed by climate driven changes to the stratification regime. The large scale warming in the northern hemisphere since the 1970s has been attributed to anthropogenic radiative forcing and also to the multidecadal pattern of the North Atlantic Oscillation

(NAO) which had a positive phase from the 1980s to the early 2000s resulting in poleward heat transport (Delworth et al., 2016). However, the next decade may see a more negative phase with lower temperatures (Delworth et al., 2016) and this




could increase the likelihood of a full overturn in these lakes which, in some cases, would transport substantial amounts of nutrients, given their accumulation in the hypolimnion, and radically alter the phytoplankton dynamics (Rogora et al., 2018; Salmaso et al., 2018).


Wind speed was included in the NPMR for lakes Garda, Como and Iseo but at generally lower sensitivity values than other variables. Over the first part of the data series higher wind speed was associated with higher chlorophyll-a while after 2010 lower wind speeds were typically related to higher chlorophyll-a. This matches the change in tendency for chlorophyll-a maxima to switch from occurring in the spring to the summer – reflecting the different wind speeds that occur at this time – higher in spring and lower in summer. The inclusion and change in influence over the time series could also reflect the different effect on the different plankton communities at these times. High winds mixing the water column usually favour diatom development during lower light conditions in spring while low winds would favour concentration of buoyant phytoplankton such as cyanobacteria towards the surface in summer (Leoni et al., 2014; Moss, 1988). While rainfall was not included directly in any of the NPMR models, the increasing frequency of extreme rainfall events have been linked to transporting nutrients to lake Maggiore supporting summer blooms of cyanobacteria that have occurred regularly since 2005 (Morabito et al., 2018).

The observation of a period of disruption to the cycle of at least two years is interesting. It may represent a lag phase when the phytoplankton community is adjusting to changed environmental conditions resulting from reducing water column mixing. Although given the significant diversity in the phytoplankton and the persistence, at least at a low density of many species it would be unusual that some species would not responded quicker (Moss, 1988). Also in the ocean, reduced mixing has been found to lead to oscillations and disruption to phytoplankton through increasing loss of nutrients via sinking phytoplankton coupled with less upward movement of nutrients from lower layers (Huisman et al., 2006). However, the reduction in production driven by ocean stratification (Behrenfeld et al., 2006) was not, in purely proportional terms, as notable as the two year drop observed here. Further work is needed to examine the changes in species composition and drivers behind this two year period of low chlorophyll-a in order to understand if there are lessons on the implications of climate change for other ecosystems. In Lake Garda the chlorophyll-a continued to be consistently lower after 2012, the reasons for which are uncertain but could be related to the fact that the temperature is consistently 1 to 2 °C higher in this lake compared to the others (Rogora et al., 2018) which could strengthen the stratification further inhibiting nutrient transfer to the epilimnion.

The increase in temperature with climate change, mediated by the NAO has led to an earlier occurrence of the spring diatom bloom in Sweden based on March NAO values (Weyhenmeyer et al., 1999). While the key climatic driver of warmer temperatures leading to earlier spring bloom may be clear, the implications posed by the absence of winter mixing may be more complex. It is expected that the species composition and abundance of the phytoplankton is likely to be altered and undergo disruption to the established seasonal succession of species. Ordinated long timeseries of phytoplankton community composition have shown shifts from being associated with euphotic depth and nutrient ratios in the 1980s to stratification strength and thermocline depth in the 1990s (Winder and Hunter, 2008). Moreover, while blooms of cyanobacteria are notably favored by increasing water temperature and column stratification in nutrient-rich lakes, blooms of different phytoplankton groups or taxa may occur in oligotrophic conditions at analogous physical conditions of high water column stability. *Mougeotia*, a filamentous green phytoplankton species that competes well at lower nutrient levels (Salmaso, 2000; Sommer, 1986) and stratified conditions, recorded mass developments during recent decades in some large sub-alpine lakes such as lakes Geneva, Garda and Maggiore (Tapolczai et al., 2015). In particular, it was observed to dominate at over 80% of total phytoplankton biomass during the summer of 2011 in Lake Maggiore, the highest value in records dating from 1984.





Filamentous green algae, smaller diatoms, cyanobacteria and other species capable of depth regulation are favoured with intensified stratification (Winder and Hunter, 2008). Similarly in Lake Garda the increased stratification resulted in lower epilimnion nutrients and was found to lead to a shift to motile species such as dinoflagellates and cryptophytes as well as changes in the cyanophyte community (Salmaso et al., 2018). Alteration of the phytoplankton community, development timing and magnitude of seasonal peaks in chlorophyll-a can have cascading effects on the ecosystem impacting zooplankton
and higher trophic levels such as fish (Winder and Sommer, 2012) .

Climate change can have a significant influence on lakes through indirect and unanticipated pathways. This study found reduced winter turnover likely exerted significant control reducing the concentration of chlorophyll-a but also considerably altered the seasonal pattern. At higher latitudes an increase in the length of growing season has been attributed not to a direct
influence of higher temperatures, but rather to an increasing proportion of rain relative to snow resulting in earlier delivery of the nutrient load to lakes (Maeda et al., 2019). While climate change may have direct physiological consequences associated with higher temperature, often the effects of the change to the structural functioning of the catchment and lake can have more powerful effects (O'Neil et al., 2012). The restructuring of ecosystems is likely to be of more importance than individual species temperature thresholds and difficult to anticipate.


Satellite observations may have proved essential in revealing these trends in chlorophyll-a concentration over time. The benefits of using remote sensing to assess chlorophyll-a are the good spatial, temporal and synoptic ability allowing several lakes to be assessed in a consistent approach (Giardino et al., 2014b). While field data and interpolation was used here to ensure complete monthly coverage, the principal pattern of a switch from a trough shape (spring peak, clear phase and
summer/autumn peak) to a unimodal shape (dominant summer concentrations) was visible from the satellite record alone. This is important to demonstrate as the approach could be used to examine alterations in plankton dynamics from climate change in unmonitored lakes globally. However it would be important to account for differing optical lake types to ensure a better relationship between reflectance and chlorophyll-a (Neil et al., 2019). An important drawback of using remote sensing is the satellites inability to provide a depth profile of the chlorophyll-a, which is well known to vary significantly with depth
– often having a maxima below 10 m with particular species adapted to live even deeper (Dokulil and Teubner, 2012; Leoni et al., 2014). This may be important given the switch to more motile species such as dinoflagellates and cryptophytes as in Lake Garda (Salmaso et al., 2018). It is possible that the lower surface chlorophyll-a in Lake Garda recorded by satellite could be an artefact and reflect a shift to motile species occurring at deeper depth.

Ideally a high resolution dataset covering temporal, spatial and depth gradients would be available to allow a comprehensive analysis. Several observations can be gathered per month by satellites, often during the summer period when cyanobacterial blooms occur thereby serving to define these events more adequately than a fixed field schedule could afford (Bresciani et al., 2020). However cloud cover often means that one or more months may not have adequate data. The approach here used a combination of including and adjusting in situ data and imputing missing values (20-31%) of data. This was necessary in
order to provide a complete dataset for analysis while also maximizing investment in state-sponsored field and satellite programs. Other approaches have aggregated many years of data to define seasonal patterns such as 10 year sliding windows (Maeda et al., 2019). The analysis was rerun for Lake Como without imputed data and indicated that the observed results and conclusions would be the same.

In Europe, the Water Framework Directive (WFD) requires that lakes are ecologically assessed using biological quality elements including phytoplankton (chlorophyll-a as well as taxonomic composition). Ecological status is measured as deviation from reference condition which varies depending on lake type and geographic region (Council of the European



Communities, 2000, 2013; Järvinen et al., 2013). The system adopted for alpine lakes included only altitude, mean depth, alkalinity and lake size (Wolfram et al., 2009), however the mixing characteristics of the lake were included as optional type
parameters in the directive (Council of the European Communities, 2000). Climate change may therefore create pressure to alter management targets and strategies as the lake typology and therefore ecological status boundaries may effectively change presenting a management dilemma between unobtainable goals and the need to protect and improve water quality (Cardoso et al., 2009). While chlorophyll-a may decline with continued stratification, superficially indicating an improvement, other biological quality elements such as macroinvertebrates in the sub-littoral and profundal zones may
deteriorate given the lower oxygen concentrations below the thermocline (Rossaro et al., 2007). Climate change may therefore test whether the intentions of the WFD to appraise the structure and functioning of freshwater ecosystems can be realized given current approaches.

Answering the question as to whether a tipping point has been passed leading to a regime shift and alternative state for the
sub-alpine lakes can risk being caught in semantics. A strict interpretation of a tipping point is one past which a previous ecological state cannot be regained, an irreversible change (Capon et al., 2015). Obviously the permanency of this depends on future climatic trends, however frequent past anomalies in the mixing behavior of lakes is likely a predictor of future regime shift with climate change (Woolway and Merchant, 2019).

**5 Conclusions**

A significant change in the seasonal pattern of chlorophyll-a occurred in the sub-alpine Italian lakes between 2003-2018. There was a shift from a dominant spring peak, clear water phase and summer peak to a weakened spring peak with more prominent summer concentrations. Non parametric multiplicative regression indicated that winter climatic variables were among the most important with higher values of the EA and NAO leading to higher temperatures likely reducing the amount
of thermal mixing in the lakes. The key driver is therefore likely to be higher winter temperatures linked to the failure since 2005/2006 of the lakes to fully mix resulting in lower nutrient concentrations in the epilimnion. The satellite estimation of chlorophyll-a and its capacity to gather a synoptic regional picture may have proved essential in revealing the changing trend over this 16 year period. The permanency of this significant change to the sub-alpine lakes will depend on climate change and inter-decadal climate drivers.

**Author contribution**

Conceptualization, Mariano Bresciani, Gary Free, Monica Pinardi, and Claudia Giardino; Data curation, Mariano Bresciani, Monica Pinardi, Gary Free, Nicola Ghirardi and Giulia Luciani; Formal analysis Gary Free; Methodology, Mariano Bresciani, Gary Free and Claudia Giardino; Supervision, Claudia Giardino; Validation, Mariano Bresciani, Nicola Ghirardi,
Giulia Luciani, and Claudia Giardino; Writing – original draft, Gary Free, Mariano Bresciani, Monica Pinardi, Rossana Caroni; Writing – review & editing, Mariano Bresciani, Monica Pinardi, Nicola Ghirardi, Giulia Luciani, Rossana Caroni and Claudia Giardino. All authors have read and agreed to the published version of the manuscript.

**Acknowledgements**

Part of the satellite processing activities was included in the FP7 GLaSS project (GA n. 313256) and H2020 EOMORES project (GA n. 730066), and in the Interreg Italy/Swiss SIMILE Project. Funding for data analysis was supported by the ESA CCI LAKES project (GA n. 40000125030/18/I-NB). We thank ARPA Lombardia and ARPA Veneto for provision of





environmental data. In situ chlorophyll-a data for Lake Maggiore were collected through the limnological campaigns funded
by the International Commission for the Protection of Waters between Italy and Switzerland (CIPAIS). We thank Giacomo
Foppiano assistance with data collation.

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
