# Peer review of "A regional evaluation of the influence of climate change on long term trends in chlorophyll-a in large Italian lakes from satellite data"

_Earth System Dynamics, 2020_

## Referee Comment (RC1) · Anonymous Referee #1 · 4 Aug 2020

**Review of the manuscript "A regional evaluation of the influence of climate change on long term trends in chlorophyll-a in large Italian lakes from satellite data", by Free et al.**

**General comments**

This manuscript is presented as an investigation on the variability of monthly chlorophyll concentration (CHL) in four subalpine lakes from 2002 to present. Possible causes of the change are studied with the end of a commercial statistical software, implementing the "NPMR" method.

I am disappointed with this manuscript. I was looking forward to reading it because I have had enough of papers making yet-another-CHL-algorithm-for-inland-waters, and I believe that the field needs to shift to climate research, so I liked the intentions of the paper. However, the methods are sometimes weak and sometimes poorly described. The conclusions might be true, but the evidence is not shown in the article. Overall, the reader has to believe the authors rather than be guided in the discovery.

My troubles with this manuscript start with the construction of the CHL time series. As MERIS was short-lived, MSI, OLI and especially OLCI are used. Data quality is different and processing is different, so inter-sensor biases will certainly appear, that will disrupt trend analysis, unless they are corrected (see for example the careful work in the CCI or Globcolour multi-sensor time series). This fact is completely ignored while it should be central to the manuscript. If there were some ground-truth data, individual sensor data could be first-order adjusted. But even in this case, when authors include in-situ data in the time series, they realize there is a bias between it and the satellite (which?), and adjust the in-situ data to match the satellite! Too bad this goes against recommendations of satellite calibration and validation, data merging and "ground-truthing".

As a side and less important comment, I have concerns regarding the compliance to open policy. The in-house software "BOMBER" is used, that is available "upon request". The statistical analysis is made using commercial software "HyperNiche" that implements the "NPMR" method. What if I want to reproduce the paper's results. While this might be acceptable for some editors and within certain groups, it is not certainly what should be promoted in modern science, that has to go towards open data and open software.

As a major result, it is claimed that the seasonality of CHL has changed, but I cannot see it anywhere. One would guess that the time series in Figure 2 do not look stable, but that is not the way to present such evidence. One has to make a seasonal decomposition, allowing for a varying seasonality. This also affects the "observed decline in Spring chlorophyll-a" (line 311).

These insufficient results are followed by a lengthy discussion in which the readers have to believe the statements instead of finding out themselves based on the presented evidence.

Overall, I can only recommend a rejection and new submission using proper methods and presentation of results as indicated.

**Specific comments**

When downloading datasets like ERA5 wind, for example, one has to precisely indicate the dataset that was downloaded, with a link to the product user manual. Otherwise, it is impossible to know the data the authors are dealing with. For example, if one says "wind", there may be many products delivering wind data from many types of sensors, time span and resolutions.

I am confused on the "model results" of Figure 4. I do not see any model data here. It is temperature, CHL and year. On the other hand, the contour plots lack a color bar and a reasonable size for the contour labels.

Figure 5: Here, the temperature (not °C) at which the maximum annual CHL occurs is plotted. As the maximum CHL value can change from year to year, it is not clear to me how to interpret this result.

The satellite regions of interest are not detailed. The reader does not now where does the data come from.

In the inset in Figure 1, the authors annexed part of Switzerland to Italy by incorporating the whole watershed to the latter, including some major towns like Locarno and Lugano. Specifically on the lakes, lake Maggiore lays partly in Switzerland.

I do not know the meaning of "inputing" data (line 131). Is it common within other communities? I assume it must be adding new data to a time series, but does it have other technical details behind? Again, simply citing an "R" package adds to the overall sensation of black box.

If one is going to cross the CHL data with some predictors, it would help to show the time series of these, or at least of those that the model determines as the most significant.

---

## Referee Comment (RC2) · Anonymous Referee #2 · 24 Oct 2020

The authors present an interesting study on the influence of climate change on chlorophyll-a change in several sub-alpine lakes observed by long-term satellite data. This is a very interesting research topic and represents a new promising direction. But I have two main concerns about this study. First is about the accuracy and the continuity of the satellites derived Chl-a data which is probably not the focus of this study but definitely provides the basis for the following analyses. Hence, I think it is still important to provide more information on the Chl-a data validation over the several sensors and the continuity between them, as the Chl-a data is not derived from a demonstrated product. The second concern is about the possible impacts other than climate change, such as anthropogenic activities or nutrient loadings. In other words, if it is possible

that the change in Chl-a pattern is caused or partly caused by other factors? If so, how to quantify the change caused by other factors? If not, why? Several detail comments are listed as follows: 1. Page 3, Line 113-116. Why to use 6SV code for the atmospheric correction for the three sensors? In my impression, there are several atmospheric correction tools that have been demonstrated to be suitable for the sensors, such as Acolite, C2RCC, and Polymer. 2. Page 3, Line 118-Line 120. How many ROIs are selected for each lake? Are those ROIs fixed and extracted from every image? 3. Page 3, Line 130. It is very confusing why the in-situ data are typically higher than that estimated data by satellite data. The estimated Chl-a data should be validated before to be fed into the analyses. 4. Page 3, Line 140. Total phosphorus data were mentioned here but why it was missed in the following NPMR analyses? 5. Section 4- Good discussion!

---

## Author Comment (AC1) · 6 Nov 2020

We would like to thank reviewer 1 for the comments intended to improve the manuscript and we are committed to improving the work and address the issues you have highlighted.

Reviewer 1 Issue/Comment I am disappointed with this manuscript. I was looking forward to reading it because I have had enough of papers making yet-another-CHL-algorithm-for-inland-waters, and I believe that the field needs to shift to climate research, so I liked the intentions of the paper. However, the methods are sometimes weak and sometimes poorly described. The conclusions might be true, but the evi-

dence is not shown in the article. Overall, the reader has to believe the authors rather than be guided in the discovery.

Reply We are sorry that you were disappointed with the manuscript but we have a lot of confidence in our data collected carefully over the years. We wish to thank you for the comments intended to help the manuscript and we are committed to improving the work and address the the issues you have highlighted.

Issue/Comment My troubles with this manuscript start with the construction of the CHL time series. As MERIS was short-lived, MSI, OLI and especially OLCI are used. Data quality is different and processing is different, so inter-sensor biases will certainly appear, that will disrupt trend analysis, unless they are corrected (see for example the careful work in the CCI or Globcolour multi-sensor time series). This fact is completely ignored while it should be central to the manuscript. If there were some ground-truth data, individual sensor data could be first-order adjusted.

Reply We do agree that sensors' characteristic might strongly impact (along with the algorithms used) on the chl-a retrieval from EO data. For this issue this study makes use of satellite-derived chl-a products for which previous studies demonstrated the accuracy of such products. Starting with MERIS, we made use of the chl-a products as obtained from the C2R algorithm. According to Odermatt 2012, C2R is successfully validated for low to intermediate chl-a concentrations (< 16 mg/m3) (Cui et al., 2010, Minghelli-Roman et al., 2011, Odermatt et al., 2010) and this product has been used by Bresciani et al. 2011 to demonstrate how MERIS-derived chl-a might support the implementation of the Water Framework Directive in European perialpine lakes. With respect to most recent sensors (i.e. OLI, MSI and OLCI onboard Landsat-8, Sentinel-2 and Sentinel-3 respectively) previous studies (Giardino et al., 2014; Bresciani et al 2018 and Cazzaniga et al (2019) showed a retrieval of chl-a comparable to field measurements (e.g. for OLCI -> MAE=0.55 mg/m3; for OLI+MSI -> MAE=0-43 mg/m3) ranging from 1 to 15 mg/m3. These results were obtained by using a sensor-independent image-processing-scheme (6SV+BOMBER) that was easily adapted to

the configurations (e.g. spectral setting) of OLI, MSI and OLCI sensors. Cazzaniga et al., 2019 also showed the agreement (r2=0.72) between MSI and OLCI for estimating chl-a according to the proposed sensor-independent image-processing-scheme.

Issue/Comment when authors include in-situ data in the time series, they realize there is a bias between it and the satellite (which?), and adjust the in-situ data to match the satellite! Too bad this goes against recommendations of satellite calibration and validation, data merging and "ground-truthing".

Reply Unfortunately, because of the large gap in the satellite record as a result of Meris failure in 2012 it was necessary to use field gathered data from local authorities. We just used the insitu data or interpolated to fill the gaps. We did not adjust the satellite data. We recognized that this is not ideal in the paper and also presented the timeseries with satellite only data (Fig. 3).

Issue/Comment As a side and less important comment, I have concerns regarding the compliance to open policy. The in-house software "BOMBER" is used, that is available "upon request".

Reply Yes, we do agree with the reviewer that is better to use open free software. In fact, for MERIS processing we made use of the C2R tool inside of the open-source toolbox BEAM. For the other sensors we used the 6SV code, which is also free and available on-line, and BOMBER. BOMBER is also free and available (both the executable file and the code), after having sent an email for requesting. We know that a software like BOMBER might be more appealing/modern if moved to open sw but for the moment we do not unfortunately have the resources for enabling such updating.

Issue/Comment The statistical analysis is made using commercial software "Hyper-Niche" that implements the "NPMR" method. What if I want to reproduce the paper's results. While this might be acceptable for some editors and within certain groups, it is not certainly what should be promoted in modern science, that has to go towards open data and open software.

Reply Unfortunately the software costs $150 ($99 student). There is not a free alternative at the moment.

Issue/Comment As a major result, it is claimed that the seasonality of CHL has changed, but I cannot see it anywhere. One would guess that the time series in Figure 2 do not look stable, but that is not the way to present such evidence. One has to make a seasonal decomposition, allowing for a varying seasonality. This also affects the "observed decline in Spring chlorophyll-a" (line 311). These insufficient results are followed by a lengthy discussion in which the readers have to believe the statements instead of finding out themselves based on the presented evidence.

Reply We are disappointed that reviewer 1 did not like the way the evidence was presented or find it convincing. We feel we provided several approaches to examine the change in seasonality. 1) The first is a clear plot of the time series of the data for all four lakes and it is stated that one can see a change from a concave pattern (well defined spring growth, decline and summer increase) to a more convex pattern with lower spring and higher summer concentrations. 2) This is followed by Nonparametric Multiplicative Regression (NPMR) models that model the time series with output contour plots that show how the pattern of chlorophyll a has changed overtime. NPMR is a flexible method that does not make assumptions about data distribution and does not require the data to be stationary and has been found to be more sensitive than autoregressive modelling. For example, Nicolaou & Constandinou (2016) as cited in the paper, replaced autoregressive modelling approaches with NPMR and found it better at revealing structural relationships on artificial and real data for neuroscience applications at Imperial College London. 3) We calculated the maximum chlorophyll-a (June-May) and plotted the temperature at which it occurred over time (Fig.5). This indicated a shift to the maximum occurring at higher temperatures reflecting the shift from spring to summer peaks. The sen slopes of two lakes were significant. However we were troubled that this was not clear enough – so we would propose to include a table, logistic regression results and text (see attached file).

Issue/Comment Overall, I can only recommend a rejection and new submission using proper methods and presentation of results as indicated.

Reply Please see the above replies detailing the many publications documenting our institutes work on chlorophyll validation over the decades. In addition, please see the reply above and supporting references for our methods. We have also added some more analysis of the seasonal changes using logistic regression, that further supports our conclusions.

Issue/Comment When downloading datasets like ERA5 wind, for example, one has to precisely indicate the dataset that was downloaded, with a link to the product user manual. Otherwise, it is impossible to know the data the authors are dealing with. For example, if one says "wind", there may be many products delivering wind data from many types of sensors, time span and resolutions. Reply Ok, we will state explicitly using the unique variable names: Climatic data were obtained from ERA5 - the fifth generation ECMWF reanalysis for the global climate and weather (monthly averaged data on single levels) (https://cds.climate.copernicus.eu/cdsapp#!/home). Data used for analysis included: 10m u-component of wind, 10m v-component of wind, total precipitation, 2m temperature (temperature of air at 2m above the surface). The wind components were used to calculate wind speed and direction. The fraction of cloud cover and specific humidity were obtained from the ERA5 monthly averaged data on pressure level, with all data documentation available online: https://confluence.ecmwf.int/display/CKB/ERA5%3A+data+documentation.

Issue/Comment I am confused on the "model results" of Figure 4. I do not see any model data here. It is temperature, CHL and year. On the other hand, the contour plots lack a color bar and a reasonable size for the contour labels.

Reply OK, thanks for noting this mix up in terminology. We will change the legend to read -": Model estimates (see Table 2) for chlorophyll-a ug l-1 (contour lines) against year and temperature oC. " ( The results of the model were presented in table 2.).

Unfortunately the contour plot only allows labeling of the contour line, while the shading is just to aid visualization between 1ug/l contours.

Issue/Comment Figure 5: Here, the temperature (not °C) at which the maximum annual CHL occurs is plotted. As the maximum CHL value can change from year to year, it is not clear to me how to interpret this result.

Reply ok, thanks for this- we will adjust from oC to temperature. The interpretation was simply that over time the temperature at which the maximum occurs has increased, tracing the shift from spring to summer maxima. - we will try to make this clearer in the manuscript.

Issue/Comment The satellite regions of interest are not detailed. The reader does not now where does the data come from.

Reply Thanks, we will list the coordinates of the locations.

Issue/Comment In the inset in Figure 1, the authors annexed part of Switzerland to Italy by incorporating the whole watershed to the latter, including some major towns like Locarno and Lugano. Specifically on the lakes, lake Maggiore lays partly in Switzerland.

Reply Yes, we agree with the reviewer. The dashed black line in the main figure referred to the Po River District/watershed which includes a portion of Switzerland. We will correct the figure in the revised manuscript.

Issue/Comment I do not know the meaning of "inputting" data (line 131). Is it common within other communities? I assume it must be adding new data to a time series, but does it have other technical details behind? Again, simply citing an "R" package adds to the overall sensation of black box.

Reply We will define the term and give more detail- it is essentially interpolating.

Issue/Comment If one is going to cross the CHL data with some predictors, it would help to show the time series of these, or at least of those that the model determines as

the most significant.

Reply We agree this would be good, but we are a little short on space but have included the Sen slope results on annual data from 1980-2018. This was carried out for the variables air temperature, winter temperature, wind speed and rain. This is probably too long to plot adequately.

Please also note the supplement to this comment:
https://esd.copernicus.org/preprints/esd-2020-56/esd-2020-56-AC1-supplement.pdf

**Supplement:**

In response to the comment from reviewer 1

"As a major result, it is claimed that the seasonality of CHL has changed, but I cannot see it anywhere. One would guess that the time series in Figure 2 do not look stable, but that is not the way to present such evidence. One has to make a seasonal decomposition, allowing for a varying seasonality. This also affects the "observed decline in Spring chlorophyll-a" (line 311). These insufficient results are followed by a lengthy discussion in which the readers have to believe the statements instead of finding out themselves based on the presented evidence."

We are disappointed that reviewer 1 did not like the way the evidence was presented or find it convincing. We feel we provided several approaches to examine the change in seasonality.

1) The first is a clear plot of the time series of the data for all four lakes and it is stated that one can see a change from a concave pattern (well defined spring growth, decline and summer increase) to a more convex pattern with lower spring and higher summer concentrations.

2) This is followed by Nonparametric Multiplicative Regression (NPMR) models that model the time series with output contour plots that show how the pattern of chlorophyll a has changed overtime. NPMR is a flexible method that does not make assumptions about data distribution and does not require the data to be stationary and has been found to be more sensitive than autoregressive modelling. For example, Nicolaou & Constandinou (2016) as cited in the paper, replaced autoregressive modelling approaches with NPMR and found it better at revealing structural relationships on artificial and real data for neuroscience applications at Imperial College London.

3) We calculated the maximum chlorophyll-a (June-May) and plotted the temperature at which it occurred over time (Fig.5). This indicated a shift to the maximum occurring at higher temperatures reflecting the shift from spring to summer peaks. The sen slopes of two lakes were significant.

4)

However we were troubled that this was not clear enough – so we would propose to include the table and text below. –

The occurrence of maximum chlorophyll-a was classified as occurring in either winter/spring or summer/autumn (Table x). During the period 2003-2008 the majority of peaks occurred in spring (brown shading) after which peaks during summer or autumn became more common (blue shading). The trends from a winter/spring to summer/autumn peak were tested for each lake using logistic regression and all slopes were positive with p ranging from 0.10 to 0.04 (Table x).

Table x Occurrence of chlorophyll-a maximum in either winter/spring (Nov-May, brown) or summer/autumn (June-October, blue) for the four lakes. Coefficient (β) and p are reported from the logistic regressions against year. Maximum values were calculated from June to May (trough to trough) the following year.

| Lake/Year | 03 | 04 | 05 | 06 | 07 | 08 | 09 | 10 | 11 | 12 | 13 | 14 | 15 | 16 | 17 | β | p |
|---|---|---|---|---|---|---|---|---|---|---|---|---|---|---|---|---|---|
| Garda | | | | | | | | | | | | | | | | 0.27 | 0.10 |
| Como | | | | | | | | | | | | | | | | 0.48 | 0.04 |
| Maggiore | | | | | | | | | | | | | | | | 0.29 | 0.08 |
| Iseo | | | | | | | | | | | | | | | | 0.41 | 0.06 |

---

## Author Comment (AC2) · 6 Nov 2020

We would like to thank reviewer 2 for the comments intended to improve the manuscript and we are committed to improving the work and address the issues you have highlighted.

Issue/Comment I have two main concerns about this study. First is about the accuracy and the continuity of the satellites derived Chl-a data which is probably not the focus of this study but definitely provides the basis for the following analyses. Hence, I think it is still important to provide more information on the Chl-a data validation over the several sensors and the continuity between them, as the Chl-a data is not derived from

a demonstrated product

Reply We agree that for confidence in the data this is a crucial foundation. We will include more detail and include published references to the calibration papers covering the different satellites used to construct the chlorophyll time-series. Copied here the reply following a similar question to reviewer 1: We do agree that sensors' characteristic might strongly impact (along with the algorithms used) on the chl-a retrieval from EO data. For this issue this study makes use of satellite-derived chl-a products for which previous studies demonstrated the accuracy of such products. Starting with MERIS, we made use of the chl-a products as obtained from the C2R algorithm. According to Odermatt 2012, C2R is successfully validated for low to intermediate chl-a concentrations (< 16 mg/m3) (Cui et al., 2010, Minghelli-Roman et al., 2011, Odermatt et al., 2010) and this product has been used by Bresciani et al. 2011 to demonstrate how MERIS-derived chl-a might support the implementation of the Water Framework Directive in European perialpine lakes. With respect to most recent sensors (i.e. OLI, MSI and OLCI onboard Landsat-8, Sentinel-2 and Sentinel-3 respectively) previous studies (Giardino et al., 2014; Bresciani et al 2018 and Cazzaniga et al (2019) showed a retrieval of chl-a comparable to field measurements (e.g. for OLCI -> MAE=0.55 mg/m3; for OLI+MSI -> MAE=0-43 mg/m3) ranging from 1 to 15 mg/m3. These results were obtained by using a sensor-independent image-processing-scheme (6SV+BOMBER) that was easily adapted to the configurations (e.g. spectral setting) of OLI, MSI and OLCI sensors. Cazzaniga et al., 2019 also showed the agreement (r2=0.72) between MSI and OLCI for estimating chl-a according to the proposed sensor-independent image-processing-scheme.

Issue/Comment The second concern is about the possible impacts other than climate change, such as anthropogenic activities or nutrient loadings. In other words, if it is possible that the change in Chl-a pattern is caused or partly caused by other factors? If so, how to quantify the change caused by other factors? If not, why?

Reply Yes this is a very valid point. In a lot of European lakes I suspect that the signal
of climate change will be obscured by eutrophication derived from catchment sourced nutrient additions or internal loading. We included total phosphorus data to check for the influence of this but it was not selected in any of the models - we can expand on this in the discussion. Other work on these lakes has found a stable trend so we just referenced the paper rather than make it a focus point. We will make an -"other factors point"- and tie it in with the fact that the models only explained a limited % of the data.

Issue/Comment Page 3, Line 113-116. Why to use 6SV code for the atmospheric correction for the three sensors? In my impression, there are several atmospheric correction tools that have been demonstrated to be suitable for the sensors, such as Acolite, C2RCC, and Polymer.

Reply Thanks for your comment, yes it is true that, like for 6SV, many codes have been developing in the last years and Acolite, C2RCC and Polymer are clear examples of such progresses. The codes are still under testing and given the continued develop-ment of the processors, including new algorithms being implemented, further analysis is needed to evaluate their performances across the large variety of inland water types. We are also evaluating such codes but for the time being, 6SV the method which is providing better results for the rather clear waters of subalpine lakes. Then we also preferred to use 6SV for its advantage of being sensor-independent code, in other words we can easily adapt the same 6SV version and atmospheric correction scheme (e.g. the use of the same origin of aerosols) to OLCI, OLI and MSI scenes.

Issue/Comment Page 3, Line 118-Line 120. How many ROIs are selected for each lake? Are those ROIs fixed and extracted from every image?

Reply The ROIs are kept consistent for the lakes- We selected one per lake in order to have the most consistent data for the time series. We will list the coordinates of the ROIs.

Issue/Comment Page 3, Line 130. It is very confusing why the in-situ data are typically higher than that estimated data by satellite data. The estimated Chl-a data should be

validated before to be fed into the analyses.

Reply Unfortunately, because of the large gap in the satellite record as a result of Meris failure in 2012 it was necessary to use field gathered data from local authorities. We just used the insitu data or interpolated to fill the gaps. We did not adjust the satellite data. We recognized that this is not ideal in the paper and also presented the time series with satellite only data (Fig. 3).

Issue/Comment Page 3, Line 140. Total phosphorus data were mentioned here but why it was missed in the following NPMR analyses?

Reply We included total phosphorus data to check for the influence of this but it was not selected in any of the models - we can expand on this in the discussion as mentioned in the point above.

Issue/Comment Section 4- Good discussion!

Reply Thanks for your comments to help improve the manuscript.